# Research Progress of Treatment Technology and Adsorption Materials for Removing Chromate in the Environment

**DOI:** 10.3390/ma16082979

**Published:** 2023-04-09

**Authors:** Tan Mao, Liyuan Lin, Xiaoting Shi, Youliang Cheng, Xueke Luo, Changqing Fang

**Affiliations:** 1College of Mechanical and Material Engineering, North China University of Technology, Beijing 100144, China; 2College of Mechanical and Precision Instrument Engineering, Xi’an University of Technology, Xi’an 710048, China

**Keywords:** environmental protection, heavy metal ion, chromate, adsorption treatment

## Abstract

Cr is used extensively in industry, so the number of Cr (VI) hazards is increasing. The effective control and removal of Cr (VI) from the environment are becoming an increasing research priority. In order to provide a more comprehensive description of the research progress of chromate adsorption materials, this paper summarizes the articles describing chromate adsorption in the past five years. It summarizes the adsorption principles, adsorbent types, and adsorption effects to provide methods and ideas to solve the chromate pollution problem further. After research, it is found that many adsorbents reduce adsorption when there is too much charge in the water. Besides, to ensure adsorption efficiency, there are problems with the formability of some materials, which impact recycling.

## 1. Introduction

With the development of modern industry, more and more heavy metals are involved in people’s production activities. As a typical heavy metal with excellent physical and chemical properties, chromium is widely used in metallurgy, electroplating, the leather industry, the textile industry, the construction industry, and refractory materials [1]. Cr is used to produce and process chromate. Due to the outdated processing technology and many enterprises wanting to reduce costs in the chromate production process [2], chromium-containing waste residues and sewage are often stacked or discharged into the environment without treatment in many developing countries, which seriously occupies and pollutes land resources and affects the use of land and groundwater resources [2]. As a widely used industrial medicine, chromium-containing salts are widely used in the printing and dyeing industry, industrial oxidant, and tanning process of the leather industry because of their excellent coloring ability, oxidation, and excellent tanning performance. Specifically, in the leather industry, because of the great demand for leather in the world every year, tanning is an essential step in leather production [3,4]. Because a large number of chromium-containing salts are used, the chromium-containing wastewater produced is very serious, which is very harmful to the environment [5].

A large amount of chromium-containing sewage is directly discharged into rivers or oceans without treatment, which seriously pollutes the ecological environment, threatens the survival of many organisms, inconveniences the lives of surrounding residents, and seriously threatens the lives and property safety of residents. The vast majority of chromium-containing waste residues are randomly stacked in the plant after the production process. Because the plant has not conducted anti-seepage treatment in the storage yard, many chromium-containing waste residues stacked in the open air enter the soil and water through rain washing, infiltration, and surface runoff [6,7]. The chromium-containing sewage cause the content of chromate in the local soil and water environment to significantly exceed standard levels. Because these are seriously exceeded, chromate will cause inflammation and even cancer [8,9,10], seriously threatening people’s health and life.

China has experienced many pollution incidents due to improper treatment of chromium: in August 2011, the chromium residue water pollution incident in Nanpan River in Yunnan Province was caused by 5000 tons of industrial chromium-containing waste residue introduced into the reservoir, resulting in the content of hexavalent chromium in the reservoir exceeding the standard by 2000 times. A large number of local livestock became sick and were killed, and the drinking water safety for local people was seriously threatened [11]. In 2003, Qingdao Hongxing Chemical Co., Ltd. was unable to deal with its chromium residue effectively, so it had to be piled up in the open. The infiltration of rainwater and the diffusion of chromium-containing substances caused groundwater pollution. Many workers who worked around the chromium mountain suffered from stomach disease, kidney disease, liver disease, and even cancer due to long-term exposure to a high-chromium environment. It was only in 2018 that the company began to carry out large-scale remediation of the chromium residues. However, the latest chromium residue treatment design scheme only covers an impervious treatment of the chromium residue accumulation site by building cement coffins similar to the Chornobyl accident and does not make the waste residue completely harmless, such as washing, reducing, adsorbing, and reusing the chromium salts, turning the waste residue into bricks and other building materials, or reducing it to metal chromium [12]. The following table shows the concentration of Cr (VI) in wastewater directly discharged by industry.

Table 1 shows the Cr (VI) concentrations in wastewater that are discharged directly from the three mainstream industrial sectors that use chromates on a large scale. As can be seen in Table 1, the highest average concentrations of Cr (VI) are found in wastewater discharged from the chromate production sector, at approximately 800 mg/L [13], from the electroplating industry, at 5–40 mg/L [14], and from the leather industry, at 6–10 mg/L [15]. The above industrial sectors produce wastewater concentrations well above the required discharge standards, exceeding EPA standards by more than 50 times and PRC standards by more than 100.

Through the above public hazard events and the table above, we can see that whether directly or indirectly, chromate pollutes the environment mainly in the form of sewage.

## 2. Status of Chromate Treatment Technology

The common methods of landfill, incineration, and direct marine treatment for pollutants easily cause secondary pollution and great environmental harm, even leading to large-scale public hazard events. These extensive and irresponsible treatments dissatisfy the current requirements. Table 2 shows chromate’s current mainstream treatment routes, and the advantages and disadvantages of various processing methods are very clear.

In addition, a comparison among other various processing methods is given. Bayer and co-workers [8] have developed the technology of composting sludge containing chromium. However, this application is limited to being used in a garden field and other fields with little contact with people to prevent the chromium in the compost from directly harming people. Moreover, the harm of excessive chromium absorbed by plants to humans and the environment must be studied. Like composting, reed bed technology is inevitably accompanied by certain biological toxicity. However, chemical or electrochemical methods such as chemical reduction precipitation, electrolytic precipitation, barium salt precipitation, and chemical coagulation are utilized to treat wastewater containing chromate. Undeniably, some of the above treatment processes are complex, and added chemicals will result in secondary pollution due to some by-products. 

Brick or asphalt making and other solidification and stabilization methods are also widely used in treating waste residues containing chromium, but they also have some shortcomings. Just like landfills, marine treatment, and making asphalt, there is also a risk of secondary pollution to the environment during transportation, storage, and solidification, which must be avoided. The involvement of high temperatures and oxidizing solid substances in the chrome slag brick-making process has significantly reduced the amount of hexavalent chromium in the slag. However, it does not guarantee that the finished product will be utterly hexavalent chromium-free. The chrome slag bricks can meet the toxicity requirements for use, but there is a potential environmental risk. Tang [18] found that brick-making could confine Cr (VI), but the confined Cr (VI) would leach in a liquid environment. The amount of Cr (VI) that leached from the brick samples varied with time and leaching method, and the Cr (VI) content of the leachate only stabilized after 20 days of leaching. He found that an alkaline environment was conducive to the re-dissolution of Cr (VI). In addition, these solidification methods only fix the chromium in the waste residue, and it is tough to recycle these resources, resulting in metal waste. In order to treat waste liquid containing chromium, a large amount of Chemical reagents are required to concentrate chromium before treatment (such as Na_2_CO_3_, NaOH, FeSO_4_, and other chemical reagents that precipitate chromate), increasing the treatment cost. In order to reduce emissions, chromium should be enriched at a low cost for recycling [19]. Therefore, the adsorption strategy is the most straightforward and efficient method for recycling chromate, and it has the advantages of reasonable cost and less secondary pollution [13,14,15].

A comprehensive comparison with other methods of treating hexavalent chromium pollutants shows the encouraging results of treating chromate in the environment by recovery and enrichment through adsorption. This method removes hexavalent chromium from the environment and reuses the enriched hexavalent chromium. Not only does it remove Cr (VI) from the environment, but because of the recycling, it also reduces the need for newly produced chromate, to a certain extent, and controls Cr (VI) pollution in both directions. As the adsorbent material can be recycled several times by desorption, the use cost is reduced.

## 3. Features and Advantages of the Principle of Adsorption

Currently, mesoporous materials and nanomaterials with a large specific surface area are used to remove chromate through physical adsorption, which occupies an important position in adsorption strategies. These standard materials include zeolite, metal oxides, hydroxide, graphene, and hydrogels. At the same time, it is also the research direction to improve the adsorption capacity for chromate through modifying and forming composites. Specifically, the adsorption rate of the latest adsorption materials for chromate in water is generally more than 90% and can even reach 100% under certain conditions [10]. The adsorption method does not produce additional by-products and generally does not need too many additional chemicals. 

Commonly, adsorption is a process of material transfer and enrichment at the two-phase interface of a heterogeneous system. In the current stage of water treatment, some solid materials with a large surface to attach the substances in water are widely used to achieve water purification. Specifically, heavy metal ions can be effectively removed from water through adsorption, which is feasible for treating most industrial wastewater. The adsorption forces mainly include intermolecular force (van der Waals force), chemical bond force, and electrostatic attraction. In addition, four adsorption methods can be used in the adsorption of chromate, these are listed below.

(1)Dispersive force adsorption

Dispersion force plays a significant role in adsorbing non-polar substances because it exists between all molecules. When the system’s temperature increases, the adsorbed material will be detached from the solid surface due to increased kinetic energy, resulting in desorption. Its adsorption material has a wide range of applications for dispersion force adsorption, but it is also accompanied by weak adsorption [20]. Cr (VI) in the sewage directly interacts with the surface of the adsorption material through the dispersion force to achieve the adsorption effect. This adsorption can occur as long as there are two substances.

(2)Electrostatic attraction adsorption

Electrostatic attraction adsorption can remove charged particles in water by electrostatic interaction between positively and negatively charged particles. This adsorption method is more efficient and accurate than sole intermolecular forces to adsorb charged particles in water, and it is hard for the adsorbate to desorb from the material’s surface. Chromate and dichromate in sewage form negatively charged ions, so the adsorbent surface with a positive charge can adsorb these negatively charged ions through electrostatic attraction [21,22,23,24]. In addition, the reductive groups (substances) on the adsorbent can reduce chromate and dichromate to Cr (III) positive ions, and these Cr (III) ions on the adsorbent surface achieve the adsorption of negative chromate and dichromate in sewage [25,26,27,28]. Therefore, the above methods for generating a positive charge on the adsorbent surface can achieve electrostatic attraction adsorption of chromate and dichromate.

For example, the magnetic chitosan particles developed by Mohammed F. Hamza et al. [24] have a large positive charge on their surface and can adsorb and remove negative Cr (VI) ions from water at pH = 2 in acidic aqueous solutions. In addition, the chitosan magnetite nanoparticles prepared by Maram H. Zahra et al. [28] not only integrated a large amount of positive charge, but also the reducing substances on them could reduce the Cr (VI) negative ions to Cr^3+^ ions, further enhancing the electrostatic adsorption capacity.

(3)Chemisorption

Chemisorption refers to the chemical reaction between the adsorbate and adsorbent forming chemical bonds and surface complexes, so the adsorbate will be anchored on the surface. The adsorption heat is as high as about 84–420 kJ/mol, equivalent to a chemical reaction’s heat. Moreover, it has strong selectivity to target substances and only forms monolayer adsorption. Chemical adsorption is relatively stable, and the adsorption process is irreversible due to the high chemical bonding force. It is noted that the chemical properties of the adsorbent greatly influence the adsorption effect [28,29]. The chemical adsorption process of Cr (VI) is because it participates in the chemical reaction. Cr (VI) can be effectively adsorbed and removed by reacting with the surface of the adsorbent material to form chemical bonds or complexes.

(4)Ion exchange adsorption

For ion exchange adsorption, the target ions are concentrated on the charged points of the adsorbent surface due to electrostatic attraction. The primary ions at these charged positions are replaced with the equivalent to the release of equal amounts of ions from the surface of the adsorbent. For chromate, there are two ion exchange adsorption modes: one is that the material itself contains the corresponding anion, which can be directly exchanged with chromate to achieve the effect of adsorption, and the second is that the reducing substance reacts with chromate, reduces it to a cation, and then performs ion exchange adsorption with the cation contained in the material. This can remove 85% of chromate in water. Moreover, it can be recycled [23,30,31,32,33]. 

Having covered adsorption, here is a brief description of the desorption treatment. Through desorption, we can recover the adsorbed material and reuse the adsorbent. Desorption is an operation in which the adsorbent is readily removed from its state by changing the environment in which it is placed. There are three general types: (1) reducing the pressure or concentration, (2) increasing the temperature, and (3) using chemicals.

These desorption operations have the advantage of recovering the adsorbent, of which: (1) is commonly used in the gas phase and, therefore, will not be repeated; (2) the desorption operation is called heated desorption and is used to recover adsorbent that has been used in a solvent that can withstand high temperatures. However, this method is ineffective in recovering the adsorbent; (3) it is mainly used for liquid phase adsorption. Desorption is carried out by changing the pH with chemicals or using solvents to reduce the affinity of the adsorbent + has already adsorbed a significant amount of the target contaminant. Desorption operations are carried out by using acids or bases to change the solution’s pH. This method has the advantage of being able to recover the adsorbent effectively. In chromate adsorption, most desorption operations are carried out using chemicals. Different desorption schemes and desorption processes were chosen for each different adsorbent, and three examples will be given in this paper to elucidate the desorption after adsorption.

Lu et al. [34] found that a desorbent of 0.5 mol/L Na_2_CO_3_ + 0.5 m/L NaOH was very effective in desorbing modified zeolites, with a resolution of 96.7%. In practice, the solution contains a high concentration of chromate, effectively achieving the recovery of hexavalent chromium. The adsorbent can still be put back into use after desorption, and the adsorption rate after reuse can reach 89.1% compared to the 93.4% adsorption rate of the initial use. Daradmare Sneha et al. [35] found that for Metal-Organic Framework/Alginate Composite Beads, a chromate adsorbent, 2 mol/L HCL solution was able to achieve over 80% desorption in 10 minutes, resulting in the recovery of chromate and reuse of the adsorbent. The biomaterial cross-linked graphene oxide composite aerogel prepared by Lanlan Li et al. [36] was able to achieve high desorption rates using a 0.1 mol/L HCL solution for 10 min, with a resolution of 90%, which could be cycled six times. The adsorption capacity of the adsorbent was still 82.01 ± 1.39% of the initial adsorption capacity.

## 4. Current Status of Research and Development of Adsorbent Materials

Adsorption materials mainly include activated carbon, natural zeolite, porous cellulose, hydrotalcite, and other porous materials with a large surface area. These materials can achieve ideal adsorption effects in many cases. However, to improve the adsorption capacity and adsorption accuracy of target substances in wastewater, many artificial adsorbents such as SBA-15 (highly ordered hexagonal mesoporous silica), artificial zeolite, graphene, functional resin, and other materials have been synthesized. There are many ways to classify these different adsorption materials. In terms of chemical composition, they can be divided into inorganic adsorbents, organic adsorbents, and composite adsorbents. According to the pore size, they can be divided into microporous adsorbents, mesoporous adsorbents, and hierarchically porous adsorbents. According to the regularity degree of pore structure, they can be divided into ordered pores and irregular pores.

As the classification of adsorbent materials is complex, this paper presents and compares adsorbent materials in three categories: inorganic, organic, and composite adsorbent materials. The advantages and problems of these materials are described in terms of their adsorption effect, economic value, and compliance with green protection.

### 4.1. Inorganic Adsorbent Materials

The main types of inorganic adsorbents are as follows, mineral and metal oxide adsorbents, hydrotalcite or layered metal (hydrogen) oxide adsorbents, and carbon adsorbents. All three materials have a vast amount of space within them, and with this comes an enormous specific surface area. The substantial specific surface area ensures that they can come into complete contact with the chromate in the affluent and achieve physical adsorption of the chromate in the effluent through van der Waals forces. However, these three inorganic sorbent materials also differ in their specific details. 

The main components of mineral and metal oxide adsorbents are various metallic or non-metallic insoluble compounds. For this type of adsorbent, the ionic fraction carried on the adsorbent can exchange ions with chromates in the wastewater or undergo redox reactions, thus achieving a higher adsorption sensitivity and efficiency. Because mineral and metal oxide adsorbents’ chemical properties are usually very stable, they will not form residues in the environment through appropriate delivery and recovery. Mineral and metal oxide adsorbents are very efficient and clean chromate adsorption materials.

Guo and Zhao synthesized SBA-15 from coal fly ash [37]. The adsorption experiment shows that SBA-15 reaches the adsorption equilibrium at 25 °C, pH = 4, and 30 min, and the removal rate of chromate is 92.73% (25 °C, 60 mg/L) [6]. The SBA-15 molecular sieve in Figure 1 has a lot of pores (pore size 8~11 nm) and an enormous specific surface area (600 m^2^/g). Therefore, it has a huge space to accommodate chromate, thus achieving the purpose of adsorption.

Shalini Rajput et al. synthesized magnetic magnetite (Fe_3_O_4_) nanoparticles to remove Cr (VI) from water [38]. The results showed that the maximum adsorption of Langmuir was 34.87 mg/g at 45 °C and pH = 2. As shown in Figure 2, magnetite nanoparticles with very small particle size aggregate extremely loose and have a large specific surface area, providing a sizeable available space to adsorb and accommodate chromate. Jerinv M. et al. [39] studied the removal of chromate from water by nano-Fe_2_O_3_. Their team synthesized 13 nm hematite nanoparticles via the sol-gel method and reached adsorption equilibrium at 30 min. Abukhadra. et al. [40] found that ultrasonic rolling can easily convert kaolinite into clay nanotubes. When the equilibrium adsorption time is 120 min, the adsorption capacity of chromate can reach 91 mg/g [38]. Salimkhani Sharareh and co-workers achieved a 100% adsorption rate of chromate at pH = 3 by using Linde Type A (LTA) Na-zeolite (LTA (NA)) as the absorbent [10]. Figure 3 illustrates the crystal structure of the LTA molecular sieve, where blue is silicon aluminum tetrahedrons and yellow are sodium ions. For this structure, sodium ions can be replaced by hexavalent chromium ions. As shown in Figure 3b, LTA (NA) particles are interwoven, spherical, and have a large specific surface area, ensuring they can fully contact and adsorb chromate/dichromate in water. In addition, the pore diameter distribution of LTA zeolite shows hierarchically porous characteristics (as shown in Figure 3c), and the adsorption capacity of LTA to chromate decreases with increasing temperature (as shown in Figure 3d).

As shown in Figure 4, hydrotalcite or layered metal (hydrogen) oxide adsorbent is a typical favorable octahedral structure, and the negative charges between layers balance the excess positive charges due to partial substitution of divalent or trivalent cations. The basic structure of LDHs is a metal—(hydrogen) oxygen octahedron [39]. The center of the octahedron is a metal cation, and the six top angles are OH-; the material path is composed of a two-dimensional extended layer-by-layer octahedral structure layer. Each time, it is combined by hydrogen bonding. There are crystalline water and anions between the lamellar structures, and the crystalline water can be removed without destroying the structure. Such adsorbents have a huge specific surface area and excellent lamellar structure distribution, and these two points determine that they can contain many ions. These charged particles provide a binding force for the attachment of chromate, and the lamellar structure can accommodate many chromates.

Yin et al. [41] prepared Mg-Al-layered bimetallic hydroxides (LDHs). The results showed that the adsorption capacity could reach 68 mg/g at 25 °C and 120 min, and the adsorption removal rate of Cr (VI) in wastewater could reach 95%. 

Advanced carbon materials such as activated carbon, carbon nanotubes, graphene, and other porous carbon materials have a vast surface area and excellent pore structure and other characteristics. Many carbon materials have been widely used as adsorbents for water treatment [42,43,44], specifically, high-performance adsorption carbon materials represented by high-new carbon materials and modified carbon materials.

Chen et al. [45] studied the adsorption mechanism of chromate on modified multi-walled carbon nanotubes. They used hypochlorite to modify the multi-walled carbon nanotubes and found that the adsorption capacity of modified carbon nanotubes for chromate reached 1.038 mg/g. Zhao et al. [46] conducted a study on the removal of chromate from water using modified activated carbon fiber mats, and their team used the technique of electro-adsorption; the activated carbon felt was modified with 20% nitric acid so that the removal rate of chromate in water was 94% and it had an excellent renewable performance. As can be seen in Figure 5, the fibers in the modified nanocarbon felt are interlaced to form a fine network structure. This structure is conducive to the complete contact of chromate in water to achieve an excellent chromate adsorption removal effect. Traditional lignite also has the potential to be prepared as a chromate adsorbent. Behnam Dovlati et al. [47] used weathered lignite to study the adsorption of chromate and found that weathered lignite can adsorb chromate more effectively in a short time. Therefore, it is a potential heavy metal ion adsorption material. Activated carbon, as a traditional adsorption material, has also been studied by the academic community. Wang et al. found that the maximum adsorption capacity of activated carbon for chromate can reach 19.305 mg/g at 25 °C [48].

Each of these three materials also has its characteristics in terms of preparation and modification: Figure 6 shows the preparation process of magnetite nanoparticles, and the hydrothermal method is used to synthesize the target material. This kind of material is mainly based on common metal salts as raw materials and is prepared by co-precipitation or sol-gel methods. Shalini Rajput et al. [38] used FeCl_2_ and FeCl_3_ as raw materials, tetraethylammonium hydroxide as a precipitant, and HNO_3_ to adjust the pH. The materials chosen for the experiments have very few by-products, and the resulting Fe_3_O_4_ is non-toxic and environmentally friendly and can be reused many times. It fits well with the development of green synthesis. Jerinv M. et al. [39] used Fe(NO_3_)_3_·9H_2_O and citric acid monohydrate (C_6_H_8_O_7_·H_2_O) as the raw materials for the preparation of iron oxide nanoparticles using the sol-gel method. Both the preparation process and the product are green and non-toxic to the environment.

In general, there are four methods to prepare this material: (1) co-precipitation, (2) ion exchange, (3) induced hydrolysis, and (4) calcination reduction. The co-precipitation method is essentially the same as the preparation of mineral and metal oxide adsorbents [41,49,50,51].

(1) method obtains the required precipitate by controlling the soluble salt mixture solution of divalent metal ions or trivalent metal ions under alkaline substances at a certain pH and temperature. The morphology of metal (hydrogen) oxide adsorbent ions prepared by this method depends on the supersaturation of the solution at the time of synthesis.

(2) method is to prepare layered metal (hydrogen) oxide adsorbents with different layer spacing, inorganic ions or organic ions can be selected as the support, and the required layered metal (hydrogen) oxide adsorbents with large layer spacing can be synthesized from themselves by exchange. This method can preserve its original structure and assemble and design the number and types of anions in the layers.

(3) method is the precipitation of bimetallic hydroxide in the mixed salt solution under the induced hydrolysis of divalent cations. In this process, the precipitant is continuously added dropwise to the mixture to ensure that the pH of the solution does not change to obtain the target substance. The basic process is divided into two steps. First, under the premise of a specific pH value, the hydroxide precipitation of trivalent metal cations is prepared; then, at this pH, divalent metal cations are added to the trivalent hydroxide suspension to induce hydrolysis and form a double hydroxide precipitate. It should be noted that during this period, the pH of the solution must be controlled to remain unchanged.

(4) method is divided into the following steps: 1. Configure the required proportion of metal ion solution; 2. Adjust the temperature and pH to produce precipitation from this solution; 3. Wash to obtain a clean residue; 4.Calcine at high temperature. The desired adsorbent material is then obtained.

The idea of the modification of carbon materials differs markedly from the first two: the mainstream research directions are: first, directly use carbon materials themselves for adsorption, such as weathered lignite or activated carbon, to directly adsorb chromate; the other is to chemically modify existing carbon materials, such as activated carbon, carbon nanotubes or carbon fibers, and then adsorb chromate.

Table 3 shows how new mesoporous mineral and metal oxide adsorbents with a regular pore structure, such as SBA-15, and modified multi-walled carbon nanotube and hydrotalcite or layered metal (hydrogen) oxide adsorbents with a homogeneous lamellar structure, such as LDHs, can achieve good adsorption results within acceptable adsorption times. Layered metal (hydrogen) oxide adsorbents with a homogeneous lamellar structure can also achieve good adsorption results within a sufficient time. Mineral and metal oxide adsorbents and hydrotalcite or layered metal (hydrogen) oxide adsorbents can be used to control the pore size uniformity of the synthesized material by modulating the ionic fraction or by controlling the pore size of the synthesized material. The pore size uniformity and size of the pores can be directed to achieve better chromate adsorption. Hydrotalcite or layered metal (hydrogen) oxide adsorbent, represented by LDHs, is a good balance between adsorption performance, process cost, and environmental friendliness and is worthy of further study in the field of chromate adsorption.

### 4.2. Polymeric Adsorbent Materials

Polymeric materials, specifically functional polymeric materials that can be greenly synthesized, are increasingly becoming a hot research topic as people pay more attention to the environment. Therefore, polymeric adsorbent materials have also been widely studied and explored. Organic adsorbent materials are divided into natural organic polymeric adsorbent materials and synthetic polymeric adsorbent materials, depending on the source of raw materials.

Natural polymeric materials, also known as biomass, have been widely used for a long time. Biomass materials have attracted much attention recently because of their high environmental affinity, degradable recycling, excellent hole structure, low cost, and ease of use. With the rise of the concept of green synthesis, more and more scientists have begun to use biomass materials to prepare new green adsorption materials.

As a biomaterial with various excellent biological functions, chitosan has attracted widespread attention as it has many -NH_2_ and -OH groups on its surface, which are very suitable for the adsorption of metal ions. Moreover, chitosan is natural, non-toxic, and harmless to the environment, which is very much in line with the current trend of green research and application. It is widely used in the field of adsorption. Many research teams have made considerable progress in the adsorption of chromate on chitosan because it can form a material with excellent pore structure and has many functional groups at the adsorption and modification potentials after combining it with structural resins. Its main form is a cross-linked mesh structure.

At the same time as a kind of familiar and excellent polysaccharide, cellulose has been widely used in biomedicine, chemical engineering, and adsorption. Because of their rich cellulose, many agricultural straws and products can also be used as excellent chromate adsorption materials. Its main form is a crossed reticulation.

Both play a similar role in the sorbent material, acting as structural support and adsorbent chromate. Both have steps in the extraction process, such as filtration, drying, grinding, bleaching, bleaching, and drying. However, because they are not ideal when applied directly, they are commonly subjected to chemical modification or compounding with other materials before being applied to adsorption. Chitosan is more involved in the chemical processes and cellulose is more concerned with extracting the cellulose fraction.

Most improved methods for chitosan materials are through specific chemical treatments, such as putting chitosan, modifier, and crosslinker into a beaker to mix and soak and waiting for the reaction of the chitosan surface functional groups and drugs. After the reaction, it is taken out of the sample and washed and dried. During this, the temperature and pH of the reaction solution need to be controlled [52].

Li et al. [53] prepared the QCS-PS anion exchange membrane with chitosan as the raw material to adsorb chromate. The results showed that the adsorption amount could reach 4.91 mg/g; Wang et al. [54] found that when the cross-linked chitosan was subjected to two-hour oscillations at pH = 3, the first adsorption removal rate can reach 99.1%. The chitosan/spirulina blend membrane adsorbent was developed by Rafael Gerhardt et al. [55]. This green synthetic material uses Penaeus brasilienesis as the raw material to prepare chitosan. For the chitin extraction, the shrimp waste went through demineralization, deproteinization, deodorization, and drying. Chitosan was obtained by chitin chemical deacetylation. Then, the cultured microalgae were used as reinforcements to prepare composite materials. This green synthesis material adsorbs chromate mainly through electrostatic interactions with a capacity of 35.8 mg/g. A novel bifunctional chitosan derivative developed by Francine Tatiane Rezende de Almeida et al. [56] with amphoteric ion properties has an adsorption capacity of 1.91 mmol/g in chromate adsorption experiments and has good recycling performance. The functionalization of magnetic chitosan microparticles developed by Mohammed F. Hamza et al. [24] achieved an adsorption capacity of 6 mmol/g, and the adsorption rate did not decrease by more than 6% after several cycles. The novel adsorbent based on chitosan magnetite nano-particles developed by Maram H. Zahra et al. [25] reached 5.7 mmol/g in visible light and up to 6.8 mmol/g in UV light after functionalization.

At the same time, many scientists have researched and explored cellulosic ground adsorption. Li et al. [57] studied the adsorption of chromate on acid-treated sunflower straw powder. They found that the removal rate of chromate was 99.5% at 8 g/L, 25 °C, pH = 6, and grain size of 80–100 mesh as a kind of fiber-rich material; garlic seedling leaves were also used to explore the adsorption of chromate in wastewater. Chen et al. [58] found that under the condition of pH = 5, the equilibrium adsorption time was 2 h, and the maximum removal rate of chromate was 94.91% for 40 mg/L solution. Sokeland WP [59], Meng [60], Nor Shilawati Ehishan [61], Abia Daoudal [62], and others have all studied the adsorption of chromate from water by various crop stalks and products. It is found that the adsorption of chromate in water has good results. Moreover, because of the convenience of raw materials and the complexity of the prefabrication process, it has good research and application prospects in the biomass treatment of chromate pollution in water.

Not only are natural polymers used for chromate adsorption, but a variety of human-made polymers are also used for chromate adsorption. Gel materials such as acrylics, resins, and conductive polymers (CPs) were used to study the feasibility of chromate adsorption.

(1) Acrylics can form gels with water as the dispersion medium. It can introduce hydrophobic groups into the water-soluble polymer with the cross-linked structure to form a cross-linked polymer that can expand when encountering water. This soft texture can also maintain a specific shape and absorb many solutions. Acrylics can fully expand without dissolving in water and are widely used for the adsorption of harmful substances in water because of their easy modification and the integration of a large number of binding sites that interact with pollutants in the sewage; specifically, many metal ions [63,64].

In general, a selection of monomers and crosslinkers is required to enable the hydrogel to be formed smoothly. After preparation, an ice and water bath is also required. Some hydrogels’ curing process requires controlling the solution’s pH and removing dissolved oxygen from the water. In addition, a curing agent is required for the re-curing cross-linking process to ensure it is formed.

The hydrogel prepared by Lv et al. [65] using polyacrylic acid has good adsorption sensitivity and accuracy for chromate in water, and the removal rate of trace chromate has reached 26.9%. It can be seen in Figure 7 and Figure 8 that there is a sizeable, cross-linked structure inside the hydrogel material, which has considerable space to accommodate chromate in water and fully adsorb it. Bhawna Sharma et al. [66] adsorbed heavy metal ions (Hg(II) and Cr (VI)) in water by the microwave-assisted rapid synthesis of reduced graphene oxide glycine hydrogel nanocomposites. After the research, the adsorption capacity of the composite hydrogel prepared by Bhawna Sharma et al. [67] on chromate in water reached 473.9 mg/g.

(2) Resin is also a very mature application of a wide range of materials. Considerable amounts of resin materials are produced worldwide every year. The 8-hydroxyquinoline chelating resin was prepared by Zhou et al. [67] using cross-linked styrene small white spheres as raw materials, coupled with 8-Hydroxyquinaldine after nitrification, reduction, and diazotization. After that, PS-HQD resin is obtained by suction filtration and drying in the air. The adsorption of chromate was tested in the Hac-NaAc system. It was found that the best adsorption conditions were pH = 5.8 h adsorption time, the adsorption capacity was 9.68 mg/g, and the recovery was 94.3%.

(3) Conductive polymer-based nanomaterials have gained significant interest in recent years. Conductive polymers (CPs) are well known for their outstanding characteristics. CPs consist of conjugated π-bonds and offer unique electrical, optical, and physical properties. In recent years, conductive polymers and nanocomposites have been widely used in the adsorption of environmental pollutants. The following table shows some conductive polymers for chromium ion adsorption and their chemical structures.

As shown in Table 4, these conductive polymers that can be used to adsorb chromate generally have reducing groups that chemically fix chromium ions. At the same time, the network structure of the polymer can also make it have an enormous specific surface area. This provides a structural basis for physical adsorption.

Polyaniline synthesized on jute fiber surfaces for Cr (VI) removal was reported by [70,71]. At the optimum experimental conditions (pH = 3 and temperature of 20 °C), a maximum monolayer adsorption capacity of 62.9 mg/g was observed. The removal of Cr ions by polyaniline-modified materials depends on the substrate materials and the functional groups of PANI.

The adsorption mechanism is that the amino group reduces the dichromate ion. The dichromate ions are converted to Cr (III). Cr (III) is then surrounded and adsorbed by nitrogenous groups.

Table 5 shows that natural biomass materials can achieve excellent adsorption effects after appropriate chemical modification or integration with other materials. Specifically, the functionalization of magnetic chitosan micro-particles and novel adsorbents based on, in particular, chitosan magnetite nanoparticles, two green synthetic chitosan materials, have a very high adsorption capacity when integrated with other materials. Therefore, integrating chitosan as a biomacromolecule in a non-toxic and non-hazardous way with appropriate enhancers can exploit its large specific surface area for excellent van der Waals force adsorption and integrate functional modules on it to enhance its electro- and chemisorption capacity. This provides a direction for developing high-performance adsorbent materials with high adsorption capacity, low cost, and that are non-toxic to the environment.

A further direction for research and development of cellulosic materials is to enhance their adsorption capacity through chemical modification better, as many cellulosic adsorbent materials still need to achieve 90% removal rates. The sensitivity to chromate in water can be further enhanced by integrating charged groups on cellulose or groups that can form complexes with chromate to improve its adsorption capacity and accuracy.

In contrast, most human-made organic substances are more toxic than biomass materials, and there is a risk of environmental pollution. It is possible to explore using small molecules from nature to polymerize substances with porous structures. Therefore, we need to further investigate how to prepare high-performance adsorbent materials based on macromolecules or small polymerizable molecules already in the natural environment, which will reduce costs and comply with green synthesis.

### 4.3. Composite Adsorbent Materials

Composite materials are an inclusive and targeted group of materials. We can composite existing adsorption materials or materials with adsorption potential from many directions. For target substances, we can improve and composite them on reinforcements, enhance their mechanical properties to improve their useability or application range or achieve other functions.

Kong et al. [71] developed the zeolite-loaded micron zero-valent iron composite to adsorb and remove chromate from water. It was found that the zero valent micron iron was used to reduce chromate in sewage to Cr (III), and then Cr (III) was fixed on the surface of the composite to achieve a good adsorption effect. Through experiments, their team found that this composite material has a high adsorption rate, and the absorption removal rate can reach more than 77%. This composite adsorbent’s microstructure and elemental information can be seen in Figure 9. Its loose porous structure on the microstructure makes it have an enormous specific surface area and binding sites to adsorb chromate. The distribution of elements fully reflects the wide distribution of elements in the composite. The Fe_3_O_4_/zeolite/graphite composite studied by Li et al. [72] also has a good adsorption capacity (80.2%) of chromate in water, which can reach adsorption saturation at 6 h. This material has the characteristics of easy preparation, simple delivery, low cost, and high reuse rate, and has a high research potential.

Composite adsorbents are adsorbents that have been developed by combining the properties of several materials. By integrating different materials, better adsorption of chromates can be achieved. Both inorganic and organic composites, which appear in the previous section, are able to improve their performance considerably compared to a single material. This is why composite adsorbents are a very important research direction, such as inorganic-inorganic composites, organic-organic composites, organic-inorganic composites, functional-structural composites, natural-artificial composites, and other composites. An appropriate combination of materials can significantly improve the adsorption of chromate. It can also improve the range of applications of the adsorbent, such as the ability to accommodate a wider pH range, better formability, and recovery properties. In addition, necessary modifications to the matrix and reinforcement of these materials, such as functional group modification, grafting of new reactive groups, chelation of specific metal ions, etc., can further improve the adsorption accuracy and adsorption capacity of chromate adsorbent materials.

## 5. Summary and Outlook

For chromate adsorption, four adsorption mechanisms are inevitable: dispersive force, electrostatic attraction, chemisorption, and ion exchange, and the ability to integrate these four mechanisms can significantly increase the adsorption capacity and accuracy of chromate adsorption. A green and cost-effective adsorbent that does not produce secondary contamination will be able to control the amount of hexavalent chromium in the environment. It will also create high economic benefits. At present, the use of natural substances, such as mineral salts, natural polymers, and natural small molecules, to produce high-performance adsorbent materials by means of green synthesis and material compounding will be the future direction of chromate adsorbent materials.

## Figures and Tables

**Figure 1 materials-16-02979-f001:**
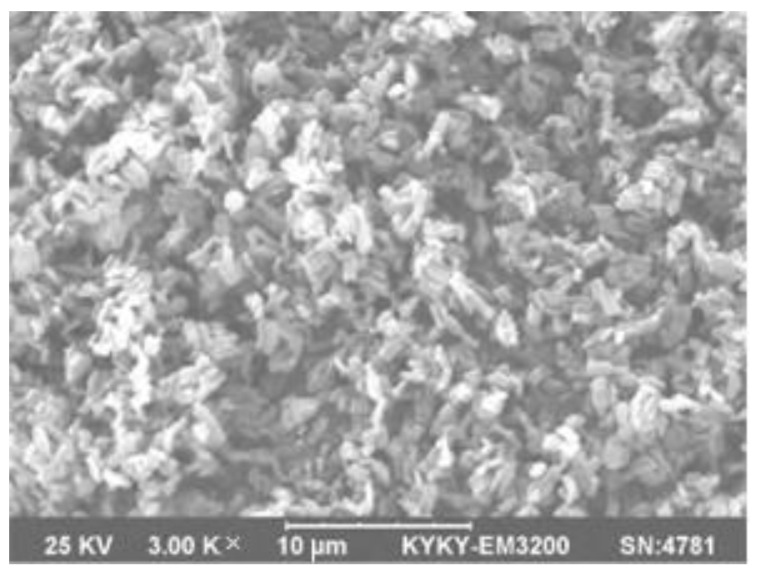
SEM image of SBA-15 molecular sieve [37].

**Figure 2 materials-16-02979-f002:**
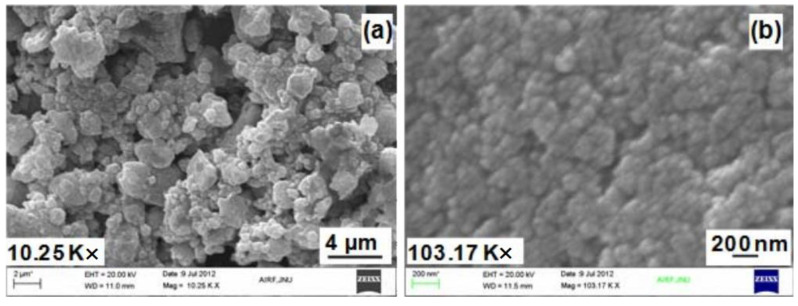
SEM micrograph of magnetite nanoparticles at (**a**) 10 K× and (**b**) 103 K× [38].

**Figure 3 materials-16-02979-f003:**
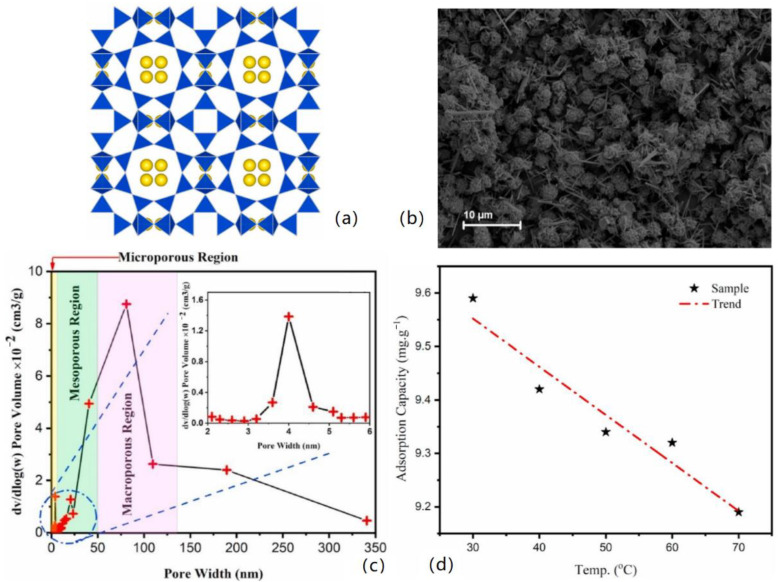
(**a**) The crystal structure of LTA zeolite is schematic. (**b**) The particle morphology of LTA (NA). (**c**) Pore width distribution of LTA zeolite. (**d**) Effect of temperature on the adsorption capacity of LTA zeolite [10].

**Figure 4 materials-16-02979-f004:**
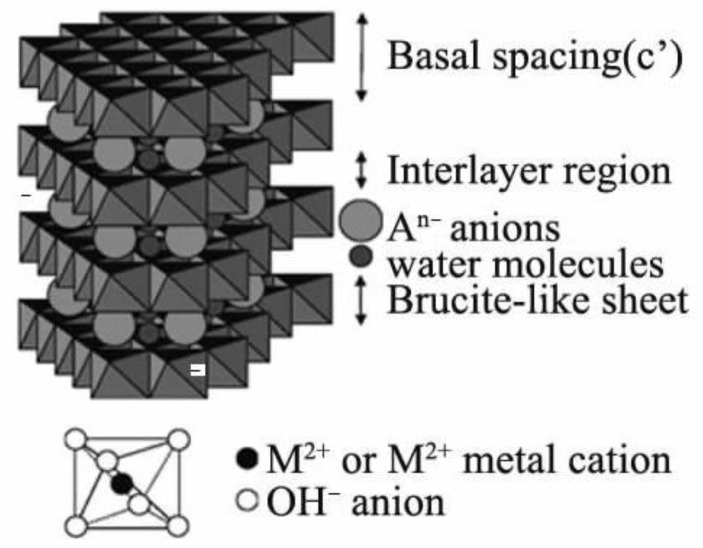
Schematic diagram of the crystal structure of LDHs (Layered Double Hydroxide) [41].

**Figure 5 materials-16-02979-f005:**
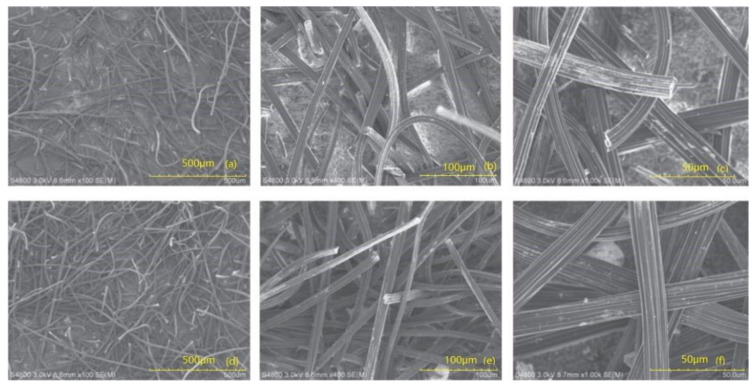
SEM images of activated carbon fiber felt before (**a**–**c**) and after modification with 20% HNO_3_ (**d**–**f**) [45].

**Figure 6 materials-16-02979-f006:**
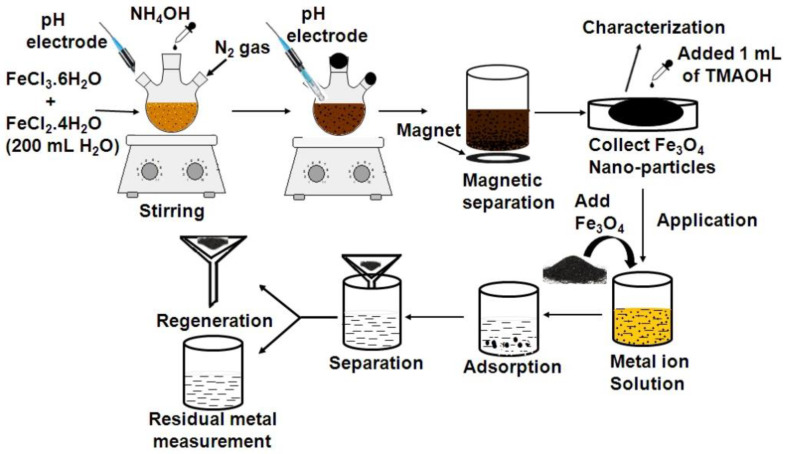
Flow chart of adsorption experiment of magnetite nanoparticles [39].

**Figure 7 materials-16-02979-f007:**
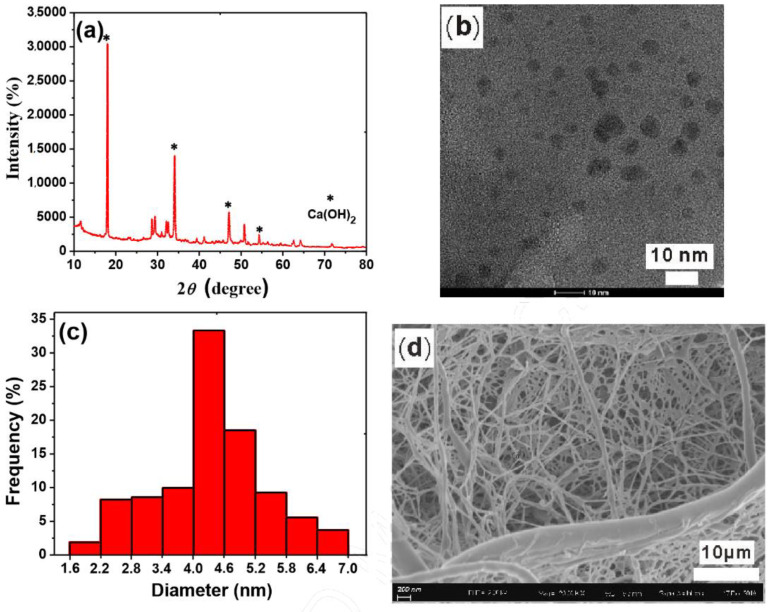
(**a**) XRD pattern for the hydration products of tricalcium silicate (Ca_3_SiO_5_), (**b**) TEM image of calcium hydroxide (Ca(OH)_2_) nano-spherulites (CNS), (**c**) histogram for TEM image of CNS with different diameters, and (**d**) SEM image of the freeze-dried swollen hydrogel cross-linked by CNS [66].

**Figure 8 materials-16-02979-f008:**
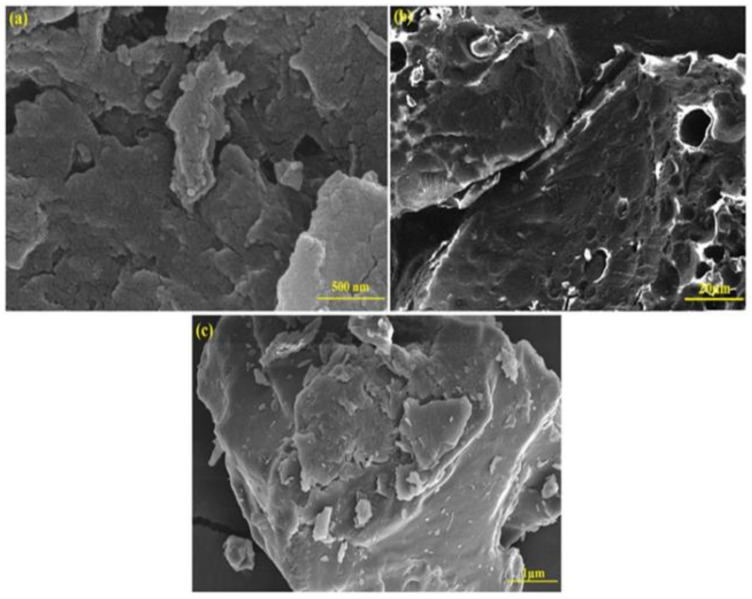
(**a**) Scanning electron microscope image of reduced graphene oxide in a reduced graphene oxide, (**b**) yellow Gel-cl-n, n-dimethylacrylamide hydrogel, and (**c**) yellow Gel-cl-n, n-dimethylacrylamide hydrogel composite [66].

**Figure 9 materials-16-02979-f009:**
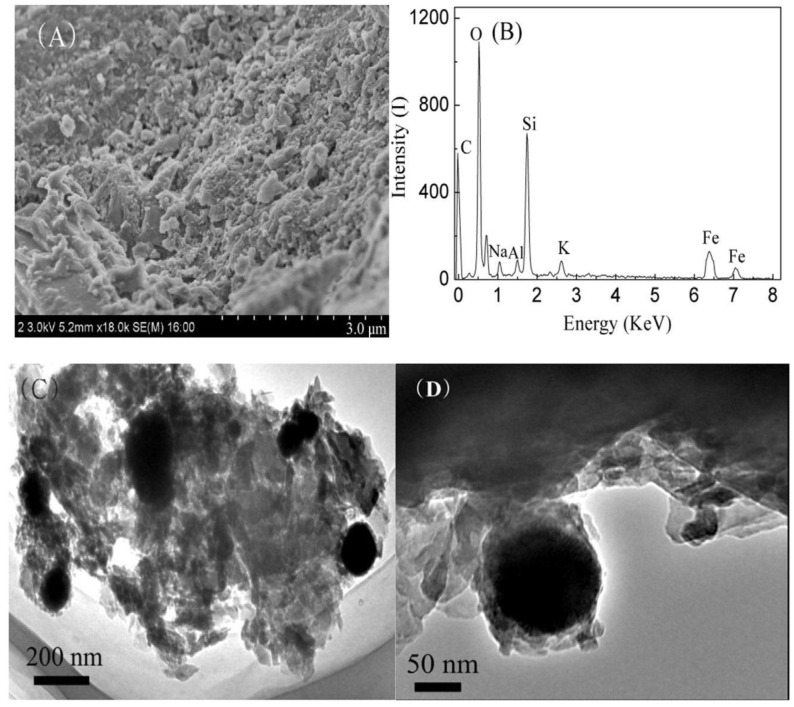
Synthesis of zeolite-supported micron grade zero-valent iron composite [71]. (**A**) SEM image; (**B**) EDX element analysis; (**C**) TEM image with 6000× magnification; (**D**) TEM image with 25,000× magnification.

**Table 1 materials-16-02979-t001:** Cr (VI) concentration in wastewater directly discharged by industry [13,14,15,16,17].

Industrial Sector	The Concentration of Cr (VI) in Sewage	Required Emission Standards
Chromate production	800 mg/L	0.05 mg/L (PRC standards) 0.1 mg/L (EPA standards)
Chromium plating	5–40 mg/L	
Leather industry	6–10 mg/L	

**Table 2 materials-16-02979-t002:** Current treatment methods for chromium-containing waste residue, sludge, and wastewater [6,15].

Method	Harm to the Environment	Product Investment	Product Quality	Economic Benefits	Comprehensive Effect of Economic Benefits
Landfill	High	Low	None	None	Normal
Incineration	Medium	Very High	None	None	Bad
Marine treatment	High	Low	None	None	Bad
Compost	Low	Low	Medium	Economic value	Good
Reed bed technology	Low	Low	None	None	Normal
Making bricks	Low	Medium	High	Economic value	Good
Making asphalt	High	High	High	Economic value	Normal
Stable curing	Low	High	None	None	Good
Adsorption treatment	None	Medium	High	Economic value	Great
Chemical treatment method	Medium	High	Medium	Economic value	Normal

**Table 3 materials-16-02979-t003:** A comprehensive comparison of representative materials [37,38,39,40,41,42,43,44].

Adsorbent	Equilibration Time	Adsorption Effect	Cost	Toxicity of Chemical Additives Used in the Synthesis Process
SBA-15	30 min	141 mg/g	High	End product is non-toxic
Magnetic magnetite	30 min	34.87 mg/g	Low	End product is non-toxic
Fe_2_O_3_ nanoparticles	30 min	Elimination of the chromate in solution	Low	End product is non-toxic
Kaolinite nanotubes	120 min	91 mg/g	Low	End product is non-toxic
LDHs	120 min	68 mg/g	Low	End product is non-toxic
Modified multi-walled carbon nanotubes	150 min	1.038 mg/g	High	End product is non-toxic
Modified activated carbon fiber mats	20 min (Electrification)	17.75 mg/g	Medium	End product is non-toxic
Activated carbon	120 min	19.305 mg/g	Low	End product is non-toxic

**Table 4 materials-16-02979-t004:** Conductive polymers for chromium ion adsorption and their chemical structures, Adapted from Refs. [68,69,70].

Polymer	Abbreviation	Structure
Polypyrrole	Ppy	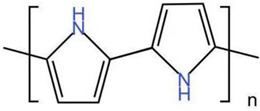
Polyaniline	PANI	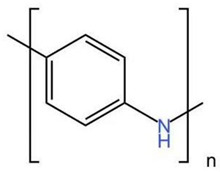
Functionalized Polyacetylene	f-PA	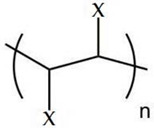

**Table 5 materials-16-02979-t005:** A comprehensive comparison of representative materials [51,52,53,54,55,56,57,58,59,60,61,62,63,64,65,66,67,68,69,70].

Adsorbent	Performance	Cost	Toxicity of Chemical Additives Used in the Synthesis Process
QCS-PS anion exchange membrane was prepared from chitosan	4.91 mg/g	Low	End product is non-toxic
Crosslinked chitosan	Under the condition of pH = 3, Cr was oscillated for 2 h, and the first adsorption rate in 99.1%	Low	End product is non-toxic
Chitosan/Spirulina blend membrane adsorbent	35.8 mg/g	Low	End product is non-toxic
Novel bifunctional chitosan derivatives	1.91 mmol/g	Low	End product is non-toxic
Chitosan nano-organics assembled adsorbent	128.43 mg/g	Low	End product is non-toxic
R6 imprinted Pebax/Chitosan/GO/Aptes nanofibers	550.5 mg/g	Low	End product is non-toxic
Functionalization of magnetic chitosan microparticles	6 mmol/g	Low	End product is non-toxic
Novel Adsorbent Based on Chitosan Magnetite Nanoparticles	5.7 mmol/g (L) 6.8 mmol/g (UV)	Low	End product is non-toxic
The tea waste	Removal rates 65%~80%	None	None
Sunflower straw powder after acid treatment	46.52 mg/g	Low	End product is non-toxic
Garlic sprout slag	3.87 mg/g	None	None
Soybean pod			
Biochar of corn straw and bagasse	Removal rates 88.74%	Low	None
Oil palm leaf adsorbent	4.564 mg/g	None	None
Thermal modified Bio-sorbent for cotton cake	40 mg/g	Low	None
Polyacrylic acid	removal rate of trace chromate has reached 26.9%		toxic
Microwave-assisted rapid synthesis of reduced graphene oxide glycine hydrogel nanocomposites	473.9 mg/g		End product is non-toxic
Polyaniline synthesized on jute fiber surfaces	62.9 mg/g	Low	toxic

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
