# Peer review of "Research Progress of Treatment Technology and Adsorption Materials for Removing Chromate in the Environment"

_materials, 2023, doi:10.3390/ma16082979_

Round 1

Reviewer 1 Report

Line 12: Chromate from water.

I feel that title needs slight change after checking the content, think.

Primary focus is on adsorbents only why?

Abstract should convey clearly what is the outcome of the work, its dragging.

Some sentences need reframing Ex: Line 25: Some amount of chromium cannot be removed.... this will be good. Similarly may I request authors to check thoroughly and reframe? Line 131: Chromium should be enriched....

Table 1: Which country standard it is ? Emission standard, mention it, if its EPA, mention it there.

Line 99: where its measured, who did? Give reference

Yale 2 harm to environment is medium in landfill is wrong, please check

Line 123-125 If brick leaches chromium  why not landfill?

Line 125 Define secondary pollution, if this is secondary, surely Landfill leaching also.

Line 130, do we have any analysis or atleast reference?

Line 133: Adsorption may be cost effective but how to dispose the adsorbed Cr? Any solution? Let us not promote pseudo, illogical science. adsorption is not a disintegration process but a removal process, in all case Cr remains intact, please think.

Total paper derailed from line 143 and authors primary focus shifted to adsorption, obviously that is non coherent with the title, please check.

Line 159: It is Spontaneous endothermic process is correct word

Line 160: If its wrong what is authors opinion, provide there.

Line 210, if other authors had already announced the works done in this area, what is this papers novelty?

I feel this paper is another repeated version of already available work with less newer inputs to a reader, author need to add some more content if possible focusing on conveying newer concepts.

Too many headings and sub headings, please optimize

Too many sub sections please optimize

Upto section 1 its just literature review, no proper interpretation exists.

Line 632: The following describes, its not brief.

Line 666-679 meaningless here, atleast in conclusion portion.

Line 716, dnt define, conclude with what is the outcome of this work

Author Response

Line 12: Chromate from water.

Has been corrected.

I feel that title needs slight change after checking the content, think.

The  title has been changed in to "Research Progress of Treatment Technology and Adsorption Materials for Removing chromate in the Environment"

Primary focus is on adsorbents only why?

The use of adsorption to treat Cr (VI) in the environment can significantly reduce the level of Cr (VI) in the environment and mitigate its harmful effects on the environment. At the same time the reversibility of the adsorption method itself allows the recovered adsorbent to undergo the corresponding desorption process for the recovery of Cr, thus saving metal resources. It is also possible to reuse the adsorbent material several times, which reduces the cost of treating Cr (VI) waste and fits in well with the green and recyclable development concept.

Abstract should convey clearly what is the outcome of the work, its dragging

The abstract has been rewritten."Cr is used extensively in industry, so the number of Cr (VI) hazards is increasing. The effective control and removal of Cr (VI) from the environment are becoming an increasing research priority. In order to provide a more comprehensive description of the research progress of chromate adsorption materials, this paper summarizes the articles describing chromate adsorption in the past five years. It summarizes the adsorption principles, adsorbent types, and adsorption effects to provide methods and ideas to solve the chromate pollution problem further. After research, it is found that many adsorbents reduce adsorption when there is too much charge in the water. According to the research, the adsorption effect of the adsorbent is reduced when there are too many charges in the water. Also, to ensure adsorption efficiency, there are problems with the formability of some materials, which impact recycling"

Some sentences need reframing Ex: Line 25: Some amount of chromium cannot be removed.... this will be good. Similarly may I request authors to check thoroughly and reframe? Line 131: Chromium should be enriched....

I corrected the English errors I could find.

Line 99: where its measured, who did? Give reference

I added the references13-15.

Table 1: Which country standard it is ? Emission standard, mention it, if its EPA, mention it there.

It was PRC standard. And I add the EPA standards this time.

Yale 2 harm to environment is medium in landfill is wrong, please check

Has been corrected into high.

Line 123-125 If brick leaches chromium  why not landfill?&9.Line 125 Define secondary pollution, if this is secondary, surely Landfill leaching also.

I added the contents as follows" Just like Landfills, Marine treatment, and Making asphalt, there is also a risk of secondary pollution to the environment during transportation, storage, and solidification, which must be avoided. The involvement of high temperatures and oxidizing solid substances in the chrome slag brick-making process has significantly reduced the amount of hexavalent chromium in the slag. However, it does not guarantee that the finished product will be utterly hexavalent chromium-free. The chrome slag bricks can meet the toxicity requirements for use, but there is a potential environmental risk. Tang.[18] found that brick-making could confound Cr(VI), but the confined Cr(VI) would leach in a liquid environment. The amount of Cr(VI) leached from the brick samples varied with time and leaching method, and the Cr(VI) content of the leachate only stabilized after 20 days of leaching. He found that the alkaline environment was conducive to the re-dissolution of Cr(VI). "

Line 130, do we have any analysis or atleast reference?

I have added the reference 19.

Adsorption may be cost effective but how to dispose the adsorbed Cr? Any solution? Let us not promote pseudo, illogical science. adsorption is not a disintegration process but a removal process, in all case Cr remains intact, please think.

I added the descriptions below," Having covered adsorption, here is a brief description of the desorption treatment. Through desorption, we can recover the adsorbed material and reuse the adsorbent. Desorption is an operation in which the adsorbent is readily removed from its state by changing the environment in which it is placed. There are three general types: (1) reducing the pressure or concentration; (2) increasing the temperature; and (3) using chemicals.

These desorption operations have the advantage of recovering the adsorbent, of which: (1) is commonly used in the gas phase and therefore will not be repeated; (2) the desorption operation is called heated desorption and is used to recover adsorbent that has been used in a solvent that can withstand high temperatures. However, this method is ineffective in recovering the adsorbent; (3) it is mainly used for liquid phase adsorption. Desorption is carried out by changing the pH with chemicals or using solvents to reduce the affinity of the adsorbent to the adsorbent. Desorption operations are carried out by using acids or bases to change the solution's pH. This method has the advantage of being able to recover the adsorbent effectively. In chromate adsorption, most desorption operations are carried out using chemicals. Different desorption schemes and desorption processes were chosen for each different adsorbent, and three examples will be given in this paper to elucidate the desorption after adsorption.

Lu et al. [34] found that a desorbent of 0.5 mol/L Na2CO3 + 0.5 m/L NaOH was very effective in desorbing modified zeolites, with a resolution of 96.7%. In practice, the solution contains a high concentration of chromate, effectively achieving the recovery of hexavalent chromium. The adsorbent can still be put back into use after desorption, and the adsorption rate after reuse can reach 89.1% compared to the 93.4% adsorption rate at the time of initial use. Daradmare Sneha et al. [35] found that for Metal-Organic Frameworks/Alginate Composite Beads, a chromate adsorbent, 2 mol/L HCL solution was able to achieve over 80% desorption in 10 minutes, resulting in the recovery of chromate and reuse of the adsorbent. The Biomaterials cross-linked graphene oxide composite aerogel prepared by Lanlan Li et al. [36] was able to achieve high desorption rates using a 0.1 mol/L HCL solution for 10 min, with a resolution of 90%, and could be cycled six times, with the recycled. The adsorption capacity of the adsorbent was still 82.01±1.39% of the initial adsorption capacity."

Line 159: It is Spontaneous endothermic process is correct word.&Line 160: If its wrong what is authors opinion, provide there.

The corresponding content has been removed after a re-investigation.

Line 210, if other authors had already announced the works done in this area, what is this papers novelty?

I feel this paper is another repeated version of already available work with less newer inputs to a reader, author need to add some more content if possible focusing on conveying newer concepts.

Too many headings and sub headings, please optimize

Too many sub sections please optimize

Up to section 1 its just literature review, no proper interpretation exists.

Line 632: The following describes, its not brief.

Line 666-679 meaningless here, atleast in conclusion portion.

Line 716, dnt define, conclude with what is the outcome of this work

Response: The structure of the article has been rearranged and questions 16-23 have been dealt with in a uniform manner.

Reviewer 2 Report

I have revised the review entitled “Research Progress of Materials for Removing Chromate in the Environment “ by Tan Mao and his colleagues. This review concerns with the possibilities techniques for removing of Cr ions. The manuscript is written well although needs some efforts to fix English language. I recommend accepted for publication after major revision.

comments

1.          The introduction section has a deficient by references; i.e., L30-46, L54-68, and L65-91 should be supported by references

2.      L130, 298, 328, 338, 443, 478, 513, 592.etc; please correct the formulas  

3.      L 149-156 should be rewritten

4.      L234-245 authors should support the information by these refences

https://doi.org/10.1016/j.cej.2021.131775, https://doi.org/10.3390/catal12070678

5.      The green synthesized materials considered as one of promising tools for recovering of Cr ions, authors should discuss this point in details to complete the work and support by references

Author Response

1.The introduction section has a deficient by references; i.e., L30-46, L54-68, and L65-91 should be supported by references

I have added the references1-5,11-12,13-15

2.L130, 298, 328, 338, 443, 478, 513, 592.etc; please correct the formulas  

Corresponding changes have been made

3.L 149-156 should be rewritten

The whole artical has been re-arranged.

4.L234-245 authors should support the information by these refences

“https://doi.org/10.1016/j.cej.2021.131775, https://doi.org/10.3390/catal12070678”

I have quoted both articles in my article.

5.The green synthesized materials considered as one of promising tools for recovering of Cr ions, authors should discuss this point in details to complete the work and support by references.

I have scattered references to the importance of green synthesis in the section of the article with synthetic materials, and in the final summary of implications, and have cited the corresponding articles.

Round 2

Reviewer 2 Report

The manuscript was improved and it can be accepted for publication in present form